# Impact of Short-Term Isoflavone Intervention in Polycystic Ovary Syndrome (PCOS) Patients on Microbiota Composition and Metagenomics

**DOI:** 10.3390/nu12061622

**Published:** 2020-06-01

**Authors:** Christoph Haudum, Lisa Lindheim, Angelo Ascani, Christian Trummer, Angela Horvath, Julia Münzker, Barbara Obermayer-Pietsch

**Affiliations:** 1Department of Internal Medicine, Division of Endocrinology and Diabetology, Medical University Graz, 8010 Graz, Austria; lisa.lindheim@gmail.com (L.L.); angelo.ascani@medunigraz.at (A.A.); christian.trummer@medunigraz.at (C.T.); Julia.Muenzker@medizin.uni-leipzig.de (J.M.); barbara.obermayer@medunigraz.at (B.O.-P.); 2Center for Biomarker Research in Medicine (CBmed), 8010 Graz, Austria; 3Department of Internal Medicine, Division of Gastroenterology and Hepatology, Medical University Graz, 8010 Graz, Austria; angela.horvath@medunigraz.at; 4Department of Medicine, Integrated Research and Treatment Centre for Adiposity Diseases, University of Leipzig, 04103 Leipzig, Germany

**Keywords:** polycystic ovary syndrome, isoflavone, equol, microbiome, androgens, glucose metabolism, metagenome

## Abstract

Background: Polycystic ovary syndrome (PCOS) affects 5–20% of women of reproductive age worldwide and is associated with disorders of glucose metabolism. Hormone and metabolic signaling may be influenced by phytoestrogens, such as isoflavones. Their endocrine effects may modify symptom penetrance in PCOS. Equol is one of the most active isoflavone metabolites, produced by intestinal bacteria, and acts as a selective estrogen receptor modulator. Method: In this interventional study of clinical and biochemical characterization, urine isoflavone levels were measured in PCOS and control women before and three days after a defined isoflavone intervention via soy milk. In this interventional study, bacterial equol production was evaluated using the log(equol: daidzein ratio) and microbiome, metabolic, and predicted metagenome analyses were performed. Results: After isoflavone intervention, predicted stool metagenomic pathways, microbial alpha diversity, and glucose homeostasis in PCOS improved resembling the profile of the control group at baseline. In the whole cohort, larger equol production was associated with lower androgen as well as fertility markers. Conclusion: The dynamics in our metabolic, microbiome, and predicted metagenomic profiles underline the importance of external phytohormones on PCOS characteristics and a potential therapeutic approach or prebiotic in the future.

## 1. Introduction

Polycystic ovary syndrome (PCOS) is one of the most prevalent endocrine and metabolic disorders in women, affecting 5–20% already from reproductive age worldwide, with phenotypes of various degrees of hyperandrogenism, anovulation and polycystic ovarian morphology [1,2].

Though the etiology is still under investigation, chronic pro-inflammatory status and a tendency toward insulin resistance [3,4,5] play a critical role in the development of harmful long-term consequences such as diabetes mellitus type 2 (T2DM) and the associated risk of pregnancy complications such as gestational diabetes [6], and increased risk of cardiovascular disease (CVD) [7]. As abnormalities in glucose metabolism of PCOS women are not yet fully understood, recent publications hint a possible connection to incretins, e.g., Glucagon-like peptide-1 (GLP-1) as well adipocyte-derived adiponectin linking glucose and fatty acid metabolism [8,9]. Incretins are metabolic hormones secreted by enteroendocrine cells, which modulate blood glucose levels by enhancing insulin secretion [10].

Lower insulin sensitivity has been found independent of BMI [11], while hyperinsulinemia and impaired glucose uptake is observed in a significant proportion of women with PCOS.

The clinical phenotype is further worsened as a result of excessive ovarian and adrenal androgen release as well as hepatic sex hormone-binding globulin (SHBG) [12] which secretion is independent of obesity [13].

Recent evidence described important differences between the gut microbiome composition of PCOS patients compared to healthy individuals, reporting a much lower diversity and an altered phylogenetic profile compared to controls [14,15,16]. The dysbiosis of gut microbiota has been shown to be largely involved in the pathogenesis of multiple systemic inflammatory diseases such: inflammatory bowel disease (IBD), multiple sclerosis (MS), systemic inflammatory arthritis, asthma, and nonalcoholic fatty liver disease, and should be considered as a novel therapeutic target of intervention [17]. A large body of literature has sought to characterize the complex interaction between foods, microbes, and derived metabolites in relation to local intestinal and systemic immune responses.

The gut microbiome, as a composition of hundreds of different bacterial strains, has proven to be capable of converting phytoestrogens, naturally occurring “xenoestrogenic” compounds from dietary sources, into hormonally active metabolites [18,19]. This effect could potentially be elicited through polyphenol compounds such as isoflavones, not only for the direct effect on microbiota composition, but also for the antiandrogenic and antioxidative microbial metabolites, with a beneficial effect in PCOS.

Isoflavones are a subgroup within flavonoid subtypes, commonly found in soy products. These naturally occurring isoflavonoids have activity similar to hormones in plants, with chemoprotective properties [20,21]. Their potential clinical effectiveness has been investigated in epidemiological reports and intervention trials [22,23,24] with beneficial outcomes proven in a broad range of different diseases [22,25,26,27]. Positive and negative regulations have been described, as there are studies hypothesizing an endocrine disruptor effect of isoflavones [28,29]. Some of these microbial strains are responsible for highly important pathways including energy adsorption, lipopolysaccharide, and short-chain-fatty acid metabolization and the bile acid pathways [30,31]. Studies on differences in the strain composition in other diseases already showed the importance of single alterations in strains on the phenotype and differences between healthy and diseased subjects. Focusing on the phenotypic effects of PCOS, e.g., inflammation and insulin resistance and finding strains that have the capacity to influence/ameliorate patient symptoms is the main goal of this study.

Daidzein and genistein, the predominating “isoflavones” in soy beans, are weak ligands for the estrogen receptor and have been shown to exert estrogen-modulating, anti-inflammatory, and antioxidant activity [19,23].

Among the isoflavones which are found in soybeans, the metabolite equol derives from the bacterial conversion of its precursor, daidzein, and proved to offer potential biological effects along with more efficient absorption and longer half-life than its precursor within the plasma [32]. However, the exact driving mechanism of this conversion and the bacteria involved remain yet unknown.

Equol binds to the estrogen receptor (ER) and G-protein-coupled estrogen receptor 1 (GPR30) which, among other hormonal effects, leads to increased intracellular Ca^2+^ levels [33]. Equol reduces the incretin effect after meals, which might contribute to its metabolic influences in the longer term [34].

The pharmacological activities deriving from isoflavone consumption may be greater in or even limited to individuals with a gut microbiota capable of adequate equol production [19]. A so-called “equol producer” has been defined as an individual producing clinically relevant amounts of equol from the precursor molecule daidzein [35]. Setchell et al. have described a method to identify equol producers using the urinary equol:daidzein ratio with a prevalence in Western populations of approximately 35% [19,36].

We hypothesized that PCOS-related gut microflora dysbiosis plays a role in equol production and effects. We assessed the changes of systemic glucose-associated parameters, metagenomic pathways, and microbiome after a short-term induced oral isoflavone intervention as described elsewhere [36] and evaluated reproductive and metabolic patterns in PCOS. Isoflavones could, therefore, be used as a prebiotic, which acts as a nutrient to support the growth of beneficial microbial strains in PCOS women or vice-versa, the use of probiotics, which are already known beneficial strains, might enhance microbial capacity to metabolize isoflavones via their properties/on their own [37].

## 2. Materials and Methods

### 2.1. Study Design

A short-time interventional pre-post study was used to examine the extent of variation in equol metabolism between a cohort of women affected by PCOS and a cohort of hormonally and metabolically healthy women. The study was conducted at the Division of Endocrinology and Diabetology, Department of Internal Medicine, Medical University Graz, Austria. The study protocol was approved by the ethical committee of the Medical University Graz (EK 26-347 ex 13/14). All participants were at least 18 years old and provided written informed consent. While maintaining their habitual lifestyle, study participants were asked to complete a three-day dietary intervention based on a previously established protocol [36].

PCOS was diagnosed according to the 2003 revised Rotterdam Criteria defined by the European Society of Human Reproduction and Embryology (ESHRE) and the American Society for Reproductive Medicine (ASRM) [38]. According to these criteria, at least two of the following needed to be present to diagnose PCOS: biochemical or clinical hyperandrogenism, oligo- or anovulation, and polycystic ovarian morphology. The three diagnostic features were assessed and related disorders were excluded [14]. The disorders adrenal hyperplasia, Cushing’s syndrome, and androgen-secreting tumors, mimic specific phenotypes of PCOS and, therefore, needed to be excluded. Only premenopausal women were included in the study. Women in the control group did not meet any of the Rotterdam Criteria, with the exception of one case of isolated long-standing mild hirsutism (Ferriman–Gallwey score of 10, no other PCOS criteria) which did not significantly alter any outcomes and is seen as a naturally occurring event in non-PCOS populations [39]. Exclusion criteria for all women were oral contraceptive, antidiabetic, or antibiotic drug use within the preceding three months, acute or chronic gastrointestinal or periodontal disease, active infections at any body site, a body mass index (BMI) <18 or >40, a known allergy to soy, and smoking (Table 1). It is obvious that these factors can influence and significantly alter not only laboratory parameters but also microbial composition. Therefore, careful selection of patients by strict inclusion/exclusion criteria is essential for reliable results later on [40,41,42].

### 2.2. Study Visits and Sampling

From 230 screened patients (Figure 1), 25 eligible women were diagnosed with PCOS by experienced clinicians and 25 metabolically healthy controls, all reported to the Endocrinological Outpatient Clinic of the Medical University of Graz in the morning after an overnight fast. Anthropometric and medical history data were obtained, and a baseline blood sample was drawn (Table 1). Of initially 50 participants, six were excluded from the analysis either due to a low BMI < 18 (*n* = 1 PCOS), previously undetected hyperandrogenemia (*n* = 3 controls), or smoking during the study period (*n* = 2 controls). One control subject did not complete the second study visit and could not be screened for equol producing capacity or included in the microbiome analysis.

Next, a 75 g oral glucose tolerance test (oGTT; Glucoral 75 Citron, Germania Pharmazeutika, Vienna, Austria) was performed, with blood sampling after 30, 60, and 120 min. A spot urine sample was provided at baseline. To evaluate dietary changes that could have an impact on the microbial gut composition, the study participants completed a food frequency questionnaire (FFQ) which has already been published [14]. The FFG was developed by dieticians of the Clinical Medical Nutrition Therapy Unit, University Clinic Graz, modified to include soy products, and to assess the intake of major food groups [43].

Following the first study visit, a defined short-term isoflavone intervention was performed as previously stated [36] in order to achieve sustained steady-state serum concentrations as confirmed by Utian et al. [27]. Study participants consumed a soy drink (Joya© Choco Soy Drink 200 mL, Mona Naturprodukte GmbH, Vienna, Austria) twice per day (morning and evening) on three consecutive days (approximately 25 mg of isoflavones per soy drink [44]). Stool samples were self-collected before the first soy drink using empty stool collection tubes with an inbuilt spatula (Praxisdienst GmbH, Longuich, Germany), stored short-term at −16 °C, and returned to the outpatient clinic on cool packs on the morning following the last soy drink. At this time, post-intervention urine and stool samples were collected. The interval between the first and second study visits was 5 ± 1.3 (mean ± standard deviation) days. From urine samples levels of daidzein, genistein, and equol before and after isoflavone intervention were measured.

### 2.3. Biochemical Measurements

Serum total testosterone, androstenedione, dehydroepiandrosterone (DHEA), DHEA sulfate (DHEAS), and dihydrotestosterone (DHT) were measured by liquid chromatography–tandem mass spectrometry as described elsewhere [14]. Urinary genistein, daidzein, and equol were measured using liquid chromatography–tandem mass spectrometry at LGC (Cambridgeshire, UK) according to a published method [45]. Serum insulin, anti-Müllerian hormone (AMH), sex hormone-binding globulin (SHBG), and insulin were measured by automated chemiluminescence immunoassay (ADVIA Centaur XP, Roche, Rotkreuz, Switzerland). Serum luteinizing hormone (LH) and follicle-stimulating hormone (FSH) were measured by enzyme-linked immunosorbent assay (DiaSource, Louvain-la-Neuve, Belgium). Plasma total cholesterol, high-density lipoprotein (HDL) cholesterol, triglycerides, glucose, and urine creatinine were measured by automated enzymatic colorimetric assay (Cobas, Roche, Germany). To evaluate the influence of our isoflavone intervention on metabolic health, we also examined changes in fasting glucose, fasting insulin, and the homeostasis model assessment for insulin resistance (HOMA2-IR) in patients (*n* = 24) and control (*n* = 19) between T0 and T1.

### 2.4. Calculation of Indices

Body mass index (BMI) was calculated as $\frac{\mathrm{Weight}\;\left(\mathrm{kg}\right)}{\mathrm{Height}\;{\left(m\right)}^{2}}$. The homeostasis model assessment for insulin resistance (HOMA2-IR) index was calculated via the HOMA calculator v. 2.2.3 provided by the Diabetes Trial Unit, University of Oxford, UK (www.dtu.ox.ac.uk/homacalculator/, last accessed 17 December 2015). The area under the curve (AUC) for glucose and insulin was calculated from the oGTT using the trapezoidal method. Free testosterone and free DHT were calculated from total testosterone/DHT and SHBG as described previously [13]. Measured urine isoflavone concentrations were normalized to urine creatinine using the formula $100\times \frac{\;\left[C\right]\mathrm{analyte}}{\left[C\right]\mathrm{creatinine}}$. Bacterial metabolism of daidzein to equol was assessed using the equol: daidzein quotient (E:D). An equol producer was defined as having a log10(E:D) >−1.5 [36].

### 2.5. Next-Generation Sequencing of Stool Samples

Total DNA was extracted from stool samples using the MagNA Pure LC DNA Isolation Kit III (Bacteria, Fungi) on the MagNA Pure Instrument (Roche, Rotkreuz, Switzerland). Stool samples were thawed partially, and an amount approximately peanut-sized was homogenized in 500 µL 1× phosphate-buffered saline (PBS), according to the manufacturer’s instructions. A 250 µL volume of diluted sample was added to 250 µL bacteria lysis buffer in a sample tube containing MagNA Lyser Green Beads (1.4 mm diameter ceramic beads, Roche). Samples were homogenized (MagNA Lyser Instrument) followed by lysozyme treatment (Roth, Karlsruhe, Germany) at 37 °C for 30 min, then proteinase K (Roche) at 60 °C for 1 h. Lysates were incubated at 95 °C for 10 min, cooled on ice for 5 min, and centrifuged at high speed. DNA was isolated by the MagNA Pure Instrument using the manufacturer’s software. The volume of the supernatant used was 100 µL. The sample was eluted in 100 µL elution buffer. PCR reaction was performed to amplify the V1-2 region of bacterial 16S rRNA gene using primers F27 (AGAGTTTGATCCTGGCTCAG) and R357 (CTGCTGCCTYCCGTA) (Eurofins Genomics, Ebersberg, Germany) and the FastStart High Fidelity PCR System, dNTPack (Roche). A 15 µL volume of the normalized pooled PCR product was used as a template for indexing PCR to introduce barcode sequences to each sample according to Kozich et al. [46]. Amplicons of the bacterial 16S rRNA were sequenced on a MiSeq desktop sequencer (Illumina, Eindhoven, The Netherlands) according to the manufacturer’s instructions.

### 2.6. Processing of Sequencing Data

Paired-end reads were joined by the fastq-join tool. Primers were removed by Cutadapt 1.6 and USEARCH 6.1 was used for reference-based chimera detection [47].

Open reference operational taxonomic unit (OTU) picking was performed with the QIIME1.9 pipeline on the inhouse galaxy server using UCLUST against the Greengenes 13.8 database [48,49,50]. When necessary, sequences were blasted in the NCBI database for further classification [51]. Clustering was performed by UCLUST [52] with a 97% sequence similarity threshold. Fasttree was used to generate a phylogenetic tree. Alpha diversity analyses were based on PD whole tree and Shannon calculated in QIIME1.9 and beta diversity using the Calypso Biomarker Discovery pipeline with our QIIME 1.9 output [53]. All OTU were filtered for abundance in at least two samples as well as an overall minimum abundance of 10.

### 2.7. Predicted Metagenome Analysis

The differences in the functional composition of the metagenome between controls (*n* = 19) and PCOS patients (*n* = 24) was predicted using PiCRUST (Phylogenetic Investigation of Communities by Reconstruction of Unobserved States) [54] and LEfSe (linear discriminant analysis effect size) [55] in combination with the QIIME1.9. PiCRUST uses 16S rRNA sequencing (Illumina, Eindhoven, The Netherlands) to predict the gene families contributing to a metagenome and Kyoto Encyclopedia of Genes and Genomes (KEGG) pathways by the identified bacteria [56]. Since we wanted to analyze the predicted metagenomic communities and the metabolic influence of a maximum number of OTUs, only singletons were removed for PiCRUST analysis. LEfSe uses the linear discrimination analysis to search for linear combinations of variables, e.g., bacteria, to separate two groups [57]. These variables are further used to calculate the effect size as the logarithmic LDA score [58].

### 2.8. Statistical Analysis and Sample Calculations

Statistical analysis was performed by SPSS Statistics 23 (IBM Inc., New York, NY, USA). All continuous data were screened for normality and equality of variance. Normally distributed data were compared using unpaired Student’s *t*-tests. Non-normally distributed data were either log-transformed, followed by parametric testing, or compared using Mann–Whitney *U* tests. Categorical data were compared using Fisher’s Exact tests. Correlations were tested using Spearman’s correlations. *Z*-scores (*Z*) were calculated from serum parameter values using the standard mean of the samples (x) and standard deviation of the samples (S) on the normal distributed data (x), $Z=\frac{x-\;\bar{x}}{S}\;$[59]. Aggregated *z*-scores were calculated from the sum of every *z*-score in the categories of “androgens” and “fertility”.

The sample size was calculated taking into account both the published observations that approximately one-third to one-half of European individuals are capable of metabolizing the isoflavone daidzein to equol, and excrete substantial amounts of equol in urine after consuming soy [60,61,62,63,64,65,66,67,68,69,70], along with the reported variation in total isoflavonoid urinary excretion of 16-fold after a high isoflavone treatment period [71]. Specifically, among equol producers, even with very modest isoflavone intake, urinary excretion of equol varies up to 600–800-fold [67,71,72], with a 1527-fold variation registered among Caucasians [61], while nonproducers excrete only trace amounts, regardless of isoflavone intake [67]. Accordingly, assuming a fold variation of 7 in urinary equol excretion (equol producers) achieved by 30% of overall study population when the fold change under the null hypothesis is 2 (nonproducers), sample sizes of 25—as previously attempted elsewhere [36]—will attain >90% detection power, through a two-sided two-sample *t*-test based on the fold, with a significance level (alpha) of 0.05. A conceivable dropout rate of 20% was considered. In the case of missing values, patients were excluded from the analysis for that variable. All data are expressed as the median and interquartile range (IQR). For the food frequency questionnaire, participant responses were assigned points based on the reported consumption frequency. The total number of points and percentages of total points in different food groups were compared between the groups.

## 3. Results

### 3.1. Patient Characteristics

Both PCOS and control groups have been extensively characterized including clinical and laboratory parameters (Table 1). Women with PCOS were generally younger than controls with a median of five years, all of them premenopausal.

We evaluated whether the well-established risk factors, according to the most recent evidence-based guidelines, were successfully described within the cohort. With the purpose of being representative of the overall population, the factors specifically altered in PCOS [73] (e.g., weight, BMI, waist circumference, lipid profiles, sex hormones, glucose levels) were tested for statistically significant difference between the groups. BMI and waist to hip ratio (WHR) were not significantly different. In the assessment of glucose and lipid metabolism, women with PCOS showed a less favorable phenotype, with higher fasting insulin, HOMA2 IR, and AUC insulin in the oGTT. Glucose tolerance itself was not impaired in women with PCOS. Women with PCOS had higher triglyceride and lower HDL cholesterol concentrations than healthy women.

As expected, women with PCOS showed abnormalities in serum levels of steroid hormones and other reproductive parameters. LH, AMH, total testosterone, androstenedione, and DHEA were significantly higher in the PCOS group than in controls. Median free testosterone and free DHT were two- to three-fold higher in women with PCOS than in control women. Women with PCOS reported hirsutism, oligo-/amenorrhea, and polycystic ovarian morphology in 46%, 71%, and 96% of cases, respectively.

Women with PCOS reported lower consumption of grains than women in the control group. There was no significant difference in any other assessed food group, including soy products.

### 3.2. Isoflavone Metabolism and Characteristics

At baseline, urine concentrations of all three compounds, daidzein, genistein, and equol, were low, reflecting the generally low consumption of isoflavones, e.g., via soy products, in the study and in comparable populations (Figure 2A) [75]. Daidzein levels were significantly higher in control women than in PCOS women at baseline (*p* = 0.036).

After three days of regular consumption of a moderate amount of soy protein (13.2 g/day), urine levels of all three compounds increased significantly in the whole cohort (*p* < 0.0001 for daidzein and genistein and *p* = 0.003 for equol). Differentiation between equol producers and nonproducers was performed according to Setchell et al., using a log10(E:D) ratio of −1.5 as the threshold (Figure 2B) [36]. The overall prevalence of equol producers was 30% (13/43) in the whole cohort (Figure 2C), and 42% (8/19) of control women were classified as equol producers, compared to 21% (5/24) of women with PCOS (Figure 2C). This difference was not statistically significant (*p* = 0.120).

### 3.3. Metabolic Changes after Isoflavone Intervention

First, we found decreased glucose (*p* = 0.01), fasting insulin (*p* < 0.01) as well as ameliorated HOMA2-IR value (*p* < 0.02) in PCOS patients, but not in control women (*p* = 0.48, *p* = 0.70, *p* = 0.72) after isoflavone intervention.

Second, women defined as equol producers demonstrated a variation of urine equol levels 5-fold larger than nonproducers with an effect size of 0.98, independently of sample size. This was associated with lower serum total and free testosterone, androstenedione, AMH, and WHR (Figure 3A). Age was significantly associated with the rise in equol after soy consumption, making it a possible confounding factor. When examining the PCOS and control groups separately, only the association of AMH with the change in equol levels remained significant (Figure 3B,C). In women with PCOS, higher baseline genistein levels were associated with a lower LH:FSH ratio (Figure 3C).

Thirdly, to further investigate the effect of an equol rise on androgenic as well as fertility markers we used standardized *z*-scores of the respective and distilled values [76,77]. These standardized *z*-scores of total and free testosterone as well as DHT and Androstenedione were summarized to the category “androgens”. LH:FSH ratio and AMH were grouped to the category “fertility”. These two categories correlated negatively with the rise of equol (−0.364, *p* = 0.021, −0.396, *p* = 0.021 accordingly) in the whole cohort also after Bonferroni correction (*p* * < 0.025). Possibly because of the small sample size, subgroups of control or PCOS women showed no further significant differences.

### 3.4. Predicted Metagenome

Our baseline (T0) evaluations showed a decreased gut microbial alpha diversity (Shannon, *p* = 0.035, PD whole tree *p* = 0.023) in PCOS women compared to controls (Figure 4).

After three days of isoflavone intervention (T1) via soy consumption, gut microbial diversity increased significantly in both groups. Although microbial diversity after intervention was significantly higher in healthy women compared to women with PCOS (Shannon *p* = 0.036, PD whole tree *p* = 0.035), diversity in the PCOS group could be increased to healthy baseline levels (Shannon *p*= 0.0528, PD whole tree *p* = 0.669).

The beta diversity using Bray–Curtis dissimilarity showed no clustering before or after the isoflavone intervention (Figure 5).

The LEfSe result for T0 (Figure 6) and T1 (Figure 7) presents the top five bacteria contributing to each phenotype. The main contributor to PCOS changed from the Oscillospira strain (186841) before intervention to Parabacteroides distasonis (1025011) after the intervention. We used the Calypso Biomarker Discovery pipeline to validate the family S24-7 as a biomarker for the control group in both T0 and T1 with an area under the ROC curve (AUC) of 0.79 and a corrected adapted *p*-value of 0.001.

*Megasphaera massiliensis* (357302 and 358887, Figure 6) contributed to the healthy phenotype. Of note, blast analysis determined 357,302 as *Megasphaera massiliensis NP3* and 358,887 showed high similarities (blastn, 99.7%) with a close relative of *NP3*, namely *DISK18*.

Metagenome techniques were used to predict differences in the functional expression of the gut microbiome in the PCOS and control groups. PiCRUST data were constructed on the rarefied OTU table. These QIIME results showed a high mean sampling depth of 70,165 ranging from 30,988 to 91,475 which were rarefied to 30,980.

From these data, we did not only find pathway changes representing typical aspects of PCOS, including carbohydrate digestion and absorption (−45.8%, *p* = 0.02) and mineral absorption (−35.4%, *p* = 0.04) [48], but also associations to new pathways involved in the biosynthesis of plant metabolites like flavonoid biosynthesis (−45.4%, *p* = 0.02), as well as carotenoid biosynthesis (−56.3%, *p* = 0.02), indicating a metagenome change with less capacity to metabolize a major class of plant secondary metabolites.

The three days of isoflavone intervention shifted each of the indicated Kyoto Encyclopedia of Genes and Genomes (KEGG) pathways in the predicted metagenome of PCOS women toward the control group features at baseline with, e.g., less carbohydrate digestion and absorption (−43.0%, *p* = 0.02), and flavonoid biosynthesis (−43.7%, *p* = 0.03). There was no significant difference of carotenoid biosynthesis (−21.5%, *p* = 0.266) as well as mineral absorption (−24.4% *p* = 0.195) between the patient and control groups after the intervention.

## 4. Discussion

This study was undertaken to investigate the impact of isoflavone supplementation on the metabolic profile and microbiota composition of PCOS patients and controls. We found an improvement in fasting glucose and insulin sensitivity in PCOS patients, but not in controls after a short-term isoflavone intervention. This may be associated with the underlying chronic low-grade inflammatory nature of PCOS, and the possible effects of isoflavonoids on inflammatory signaling pathways. Laboratory studies have proven genistein to directly inhibit insulin-stimulated IRS-1 serine phosphorylation in endothelial cells, inhibiting inflammation in an IKKβ/NF-κB-dependent manner [78]. Though structurally similar, Daidzein acts via different molecular mechanisms, restoring the TNF-α-mediated reduction of Forkhead box protein 01 (Fox01) [79]. In fact, while the protective effect of dietary phenolics was thought to be due mainly to their antioxidant properties, recent studies have shown that the metabolites—which appear in the circulation in nmol/L to low mmol/L concentrations-exert modulatory effects via selective actions of different components on the intracellular signaling cascades vital for cellular functions. Furthermore, the intracellular concentrations required to affect cell signaling pathways are considerably lower than those required to impact on antioxidant capacity [80].

While mechanistic considerations may not be advanced, all noted improvements within the PCOS cohort, particularly among equol producers, are supported by microbial and metagenomic findings. This sheds new insight upon the involvement of microbiota biotransformation activities in PCOS and the estrogenic action of equol. Indeed, equol has been shown to directly improve pancreatic insulin secretion after oral glucose uptake in vivo [81]. Furthermore, a recent in vitro study using GLUTag cells, a model of intestinal enteroendocrine L-cells, showed an inhibitory effect of equol on intestinal glucagon-like peptide-1 (GLP-1) production despite a rise in cellular Ca^2+^ levels [33]. This could be one reason why an equol/incretin/microbiome interaction may play a crucial role in insulin secretion and glucose metabolism in PCOS [82]. The positive effect of Ca^2+^ in the context of PCOS has been shown in other studies that hypothesize a potential estrogen agonistic effect [83,84,85].

While a number of publications have addressed the potential effects of soy food on various chronic diseases, isoflavones should not be equated with estrogens. Of note, circulating levels of phytohormones from modest soy consumption may even exceed endogenous estrogen levels by several orders of magnitude [86].

Although a number of experiments have been performed to correlate isoflavones and PCOS in rats, only a handful have investigated the relationship between the intake of isoflavones and PCOS in humans. These studies described improvements in plasma lipid and androgen profiles after three to six months of isoflavone interventions in PCOS women [26,87,88,89,90,91,92]. Most of these studies have already given promising results based on markers of insulin resistance, hormonal status, triglycerides, and biomarkers of oxidative stress. Studies based uniquely on genistein often revealed contrasting results, if any at all. A limitation of genistein bioavailability after oral administration may be due to its poor water solubility [93], and also its bitter taste [94] which may provoke poor study compliance. More importantly, genistein is a multifunctional compound, and may behave differently in the regulation of insulin action under different situations. Gao et al. demonstrated both its positive and negative regulations in endothelial cells [78]. In addition, most of these studies did not provide information about the gut microbiota—as presented in our study—to further characterize intestinal bacterial diversity and evaluate the relevance of microbiota biotransformation activity. Research on the hormonal effects of soy products and their endocrine effects refering to the different phases of a woman’s life from childhood, via adolescence, adult lifetime till menopause remains scarce and lacks robust studies. One meta-analysis found that isoflavone consumption reduced circulating levels of LH and FSH [86]. So, our findings of higher baseline genistein levels in PCOS patients with a lower LH:FSH ratio may support an improved pituitary background.

Among the parameters potentially influenced by the isoflavone intervention was the metabolism of HDL cholesterol. Although not being the main focus of this study, the inter-related effects of hyperandrogenism and dyslipidemia in PCOS still remain elusive. Laboratory studies on fetal androgen exposure demonstrated transgenerational effects similar to diet-induced obesity (DIO) generating a pro-inflammatory phenotype with altered immune homeostasis in metabolic organs, affecting the expression of genes involved in adipogenesis, dysregulation of hepatic cholesterol synthesis and enterohepatic circulation, overall increased biosynthesis of hydrogen peroxide and metabolism of reactive oxygen species [95,96]. The lipid pattern typical of PCOS has been identified as a risk factor for adverse pregnancy outcomes [97], and daughters of mothers with PCOS are more likely to be diagnosed with PCOS with lipid metabolic disorders [95]. A protective effect has been suggested for androgen receptor (AR) activation both in male and females [98,99] as well as a suppressive action on the generation of bioactive lipids from polyunsaturated fatty acids (PUFAs), even though a PCOS specific elevation in phosphatidylcholine (PC) was ascribed [100]

It should be noted that such evidence was sought based on testosterone deficiency models such as androgen receptor knockout mice (ARKO), and the generation of lipid decomposition products was not thoroughly examined. PUFA-containing phospholipids, such as phosphatidylcholine, are notably prone to oxidative damage, and their peroxidation results in the generation of a complex mixture of oxidized phospholipids and terminal degradation products [101]. Although they may not be directly bioactive, they form highly reactive adducts which can modify self-molecules generating structural neoepitopes (OSEs) that are recognized by receptors of the immune system, as part of housekeeping functions, and can trigger chronic inflammation [102].

Nevertheless, the therapeutic effects of isoflavones in ameliorating the lipidic profile in PCOS women have been documented through the usage of isolated genistein [87]. Our data appear to confirm an already presumed involvement of the gut microbiota in such effect, and identifies a major efficacy deriving from a specific gut microflora composition in converting isoflavone glucosides and aglycones.

While we documented effects after a short-term intervention, the duration of isoflavone consumption for potential influences is controversial. Two trials demonstrated changes in the fecal bacterial community after 16 to 25 weeks of exposure [103,104]. The study by Nakatsu et al. on postmenopausal women found bacterial variations on equol producers after already one week [105]. In our study, the predicted metagenome analyses showed a convergence of microbiome properties in PCOS women to those of healthy controls at baseline within three days of isoflavone intervention (Figure 4). Whether such an immediate effect is based on an equivalent increase in equol producing bacteria has been investigated recently [35]. 

Women with PCOS may, therefore, benefit from consuming more isoflavone-containing nutrients, regardless of their initial equol producing capacity. Ultimately, women defined as equol producers may experience direct beneficial effects from equol binding to DHT, preventing androgen receptor activation, or via a decrease in steroidogenic enzyme expression, as it has been shown in vitro and in a rat model of PCOS [32,106]. In a more indirect manner, the capacity to produce equol could be a biomarker designated to the presence of gut bacteria which exert a beneficial influence on metabolic and reproductive processes.

The overall equol producer prevalence in our study cohort was around 35%, which fits well with previous reports on Western populations (Figure 2C) [107]. Of note—though we found only a borderline significance potentially due to the limited sample size in our study—PCOS women were only half as likely to be equol producers than healthy women (Figure 2B).

Another aspect of our study supports previous findings, demonstrating variations of urinary equol concentrations following isoflavone challenge, with an effect size within our study of 0.98. This implies not only a diagnostic feature for PCOS women, but also the basis for potential pre- and/or probiotic interventions [26]. Probiotic strains with the capacity of converting isoflavones could be beneficial for women missing such trait, and prebiotic on the other hand support these strains in their role.

Our baseline (T0) evaluations (Figure 4) confirm recent publications arguing a decreased gut microbial alpha diversity in PCOS women compared to healthy controls [2,108]. These data also align with the common concept of a fast adapting diet-induced change of the microbiome [109,110,111,112]. Although PCOS patients’ alpha diversity after invention (T1) was not as high as in the controls (Figure 4), the improvement was comparable to the alpha diversity of hormonally healthy controls before intervention.

Bacteria which attributed to the control group (T0 and T1)—including *Bacteroidales* S24-7 (Figure 6 and Figure 7)—have already been described as one of the key species for binding and metabolizing starch and have been associated with homeothermic plant-eating hosts [113]. Only small differences in LEfSe were noted when comparing the control group before and after isoflavone intervention.

Women with PCOS, on the other hand, only had *Bacteroides* 186,841 and *Megasphaera* 358,887 (Figure 6 and Figure 7) in common with their LEfSe result at baseline. However, the most prominent OTU contributing to the PCOS phenotype after isoflavone intervention was classified as *Parabacteroides distasonis* (Figure 7). This strain has been shown to modulate the host metabolism by decreasing weight gain, hyperglycemia and, therefore, alleviating obesity and metabolic syndrome in mice [114] and to have the capacity to metabolize flavonoids in humans. To the best of our knowledge, we are the first to connect *Parabacteroides distasonis* to metabolic changes in PCOS patients. The microbiome of PCOS patients contains various uncultivable members of the *Bacteroides* and *Ruminococcus* genus with yet unknown function.

LEfSe results (Figure 6 and Figure 7) indicate the differentiation of PCOS patients and controls by the abundance of features identified as *Megasphaera massiliensis,* in individuals with PCOS. *Megasphaera massiliensis DISK18* has already been described for its biofilm forming capacity as well as the production of an important oxidative stress response (e.g., chaperone dnaK gene cluster, heat shock protein 33, superoxidase reductase), carbon and phosphate starvation (e.g., stringent starvation a and b, Pho H) proteins., cofactors and vitamins (e.g., vitamin B complex) [115,116]. These strains in combination with the *Parabacteroides* and *Bacteroides* strains give a good insight into the alteration of the gut microbiome of PCOS women. These strains alter important PCOS-typical pathways and can potentially worsen the phenotype in dysbiosis. Finding and supplementing these strains has already been done in the recent past and should be tested for our targets [117].

Although some of the relevant bacteria remain uncultivable to date, future approaches of microbiome research will potentially reveal their impact on the pathophysiology and specific regulations in PCOS.

Using recent advances in metagenome statistics, we were able to underpin a significant contribution of equol production on PCOS-relevant pathways, such as carbohydrate digestion and absorption, which might directly relate to the manifestation and/or modulation of clinical symptoms in PCOS.

Predicted metagenomic changes in the present study included KEGG pathways such as the “carbohydrate digestion and absorption pathways”, which define the digestion of complex carbohydrates to monosaccharides by several enzymes prior to their absorption in the small intestine [56,118]. This uptake process is linked to the potassium dependent transporter SGLT1 (sodium/glucose cotransporter 1) that induces rapid insertion and activation of GLUT2 (glucose transporter 2), which facilitates the exocytosis of glucose in the enterocytes [119,120]. These predefined KEGG “carbohydrate digestion and absorption pathways” were significantly reduced in PCOS phenotypes, as well as “flavonoid biosynthesis and mineral absorption” while “stilbenoid, diarylheptanoid, and gingerol biosynthesis” was slightly increased.

These xenoendocrine molecules have been associated with cardiovascular health, neuroprotection, and cancer prevention [121,122,123,124].

The KEGG predefined “mineral absorption pathway” on the other hand stands for bacterial properties of passive and active transport of minerals through the intestinal mucosa. This system often uses specialized transport proteins, such as vitamin D-dependent calcium-binding protein or ferritin, and might contribute to the specific features of metabolism in PCOS women [125,126,127,128].

These alterations in metabolic KEGG pathways enlighten an as yet undescribed phenomenon in PCOS, which fits well to the DOGMA (“Dysbiosis of Gut Microbiota”) hypothesis indicating a close connection of PCOS phenotype and a dysbiosis of the gut [129]. Finally, there is the aspect of physical activity and its impact on metabolic parameters. With differences of PCOS and healthy controls in lipid parameters (Table 1) and the potentially associated higher cardiovascular risk later on, we can confirm other studies [130]. With different publications stating that physical activity improves metabolic parameters including insulin resistance, blood lipids, and hormones in healthy patients [131] but also occurs in PCOS women although an intense physical workout plan could not fully reverse the PCOS phenotype [132].

The combination of sport, healthy diet, and isoflavones had a greater impact on the metabolic parameters [133,134] indicating that lifestyle is an important co-factor that should be taken into account [135,136]. The mechanism which connects isoflavones and sport is not yet understood but studies being performed in different fields including bone and cancer [24,137] highlight the importance of research in this area. This mechanism may also be beneficial for PCOS women without clinical or biochemical hyperandrogenism as these women might suffer from metabolic risk factors associated with HA, leading to increased risk of type II diabetes mellitus, obesity, hypertension, dyslipidemia, and metabolic syndrome, and may be predisposed to later sequelae of these conditions.

The limitations of our study are the pilot character and its small sample size. Furthermore, we cannot exclude that some of our results might have arisen due to the confounding factor of age, as women in the PCOS group were five years younger than control women, and age showed a significant positive correlation with equol production capacity. However, a five-year difference and a similar age range (PCOS: 22–42, Controls 25–47) should not underpin large age-related changes in premenopausal women. In summary, since the overall size of our cohort was quite small, the risk of confounding may be more pronounced and the results need to be interpreted with caution and warrant further validation in a larger setting. Although the impact of physical activity on insulin resistance and phytoestrogens is published [138] no data were collected in this study but participants were advised not to change their lifestyle.

## Figures and Tables

**Figure 1 nutrients-12-01622-f001:**
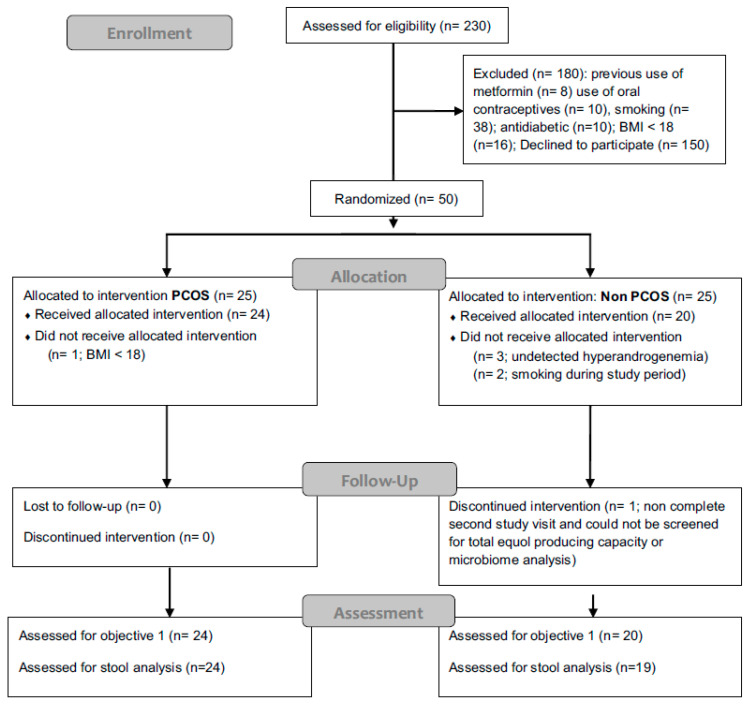
Flow diagram from patient recruitment process to the final analysis. PCOS, polycystic ovary syndrome, BMI body mass index.

**Figure 2 nutrients-12-01622-f002:**
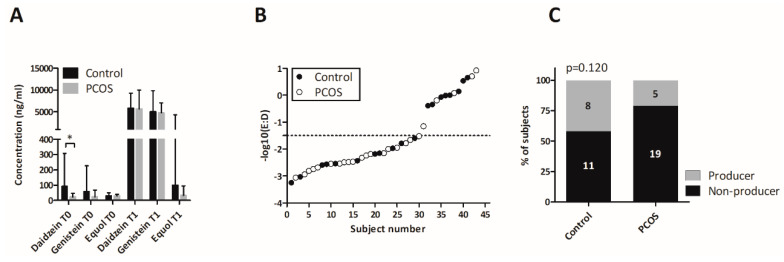
Bacterial isoflavone metabolism in women with PCOS and control women. (**A**) Daidzein, genistein, and equol levels in urine before (T0) and after (T1) a 3-day isoflavone intervention via oral soy consumption. Data are presented as median and IQR. (**B**). Determination of equol producers and nonproducers using log10(equol:daidzein). An equol producer was defined as having a quotient >−1.5. (**C**). Comparison of equol producer prevalence in PCOS and control groups. The number of subjects in each category is indicated in segments.

**Figure 3 nutrients-12-01622-f003:**
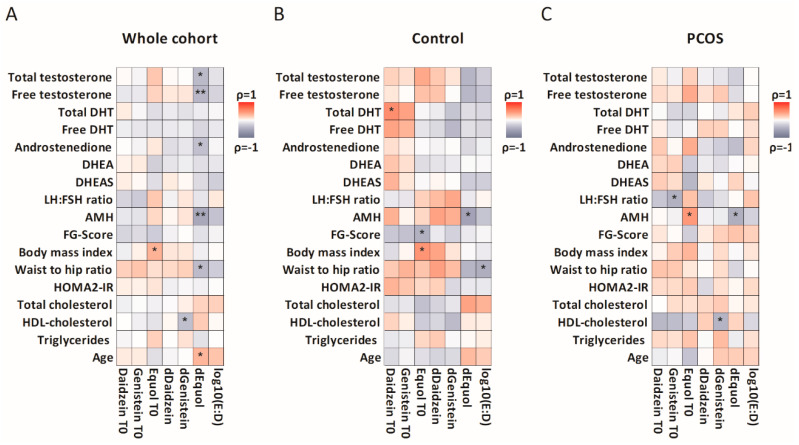
Correlations between parameters related to isoflavone metabolism, reproductive function, and glucose and lipid metabolism. (**A**). The whole study cohort. (**B**). Control women only. (**C**). Women with PCOS only. DHT, dihydrotestosterone; DHEA, dehydroepiandrosterone; DHEAS, DHEA sulfate; LH/FSH, ratio of luteinizing to follicle-stimulating hormone; AMH, anti-Müllerian hormone; FG-Score, Ferriman–Gallwey Score; HOMA2-IR, homeostasis model assessment for insulin resistance; HDL, high-density lipoprotein; T0, baseline value; d, change from baseline to T1 after isoflavone intervention. Cells are colored according to Spearman’s ρ. Significant correlations are marked as follows: * *p* < 0.05; ** *p* < 0.01.

**Figure 4 nutrients-12-01622-f004:**
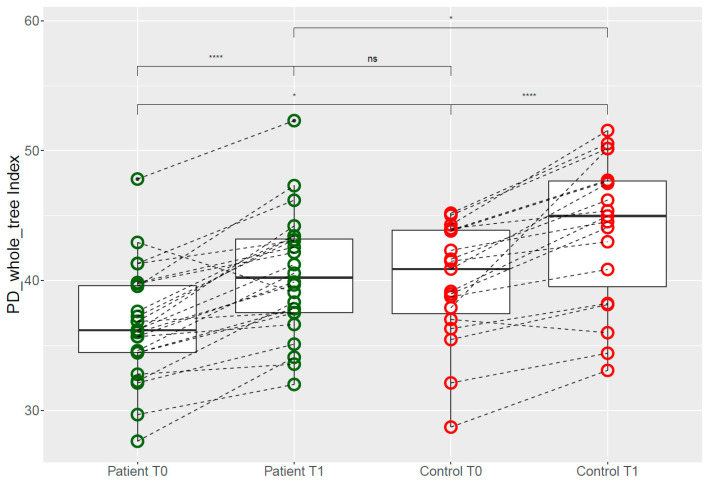
Alpha diversity change of controls and patients before (T0) and after (T1) short-term oral isoflavone intervention. Both the differences of patients (** *p* = 0.005) and controls (* *p* = 0.02) between T0 and T1 were significant using Student’s *t*-tests. There was a significant difference in diversity between controls and PCOS patients in both T0 (* *p* = 0.023) and T1 (* *p* = 0.035). However, there was no difference between controls at baseline (T0) and patients after the intervention (T1, ns, *p* = 0.669). ns *p* > 0.05, * *p* < 0.05; ** *p* < 0.01; *** *p* < 0.001.

**Figure 5 nutrients-12-01622-f005:**
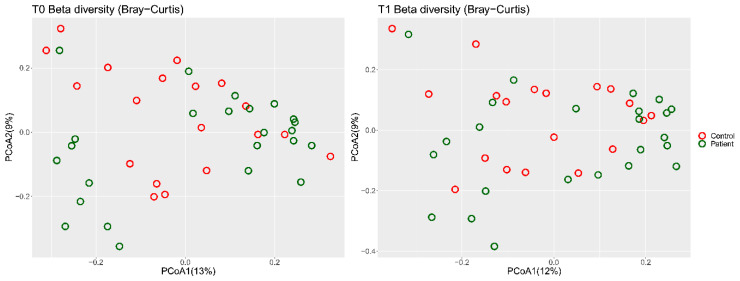
The beta diversity (Bray–Curtis) depicted no clustering before (T0) and after (T1) the short-term isoflavone intervention.

**Figure 6 nutrients-12-01622-f006:**
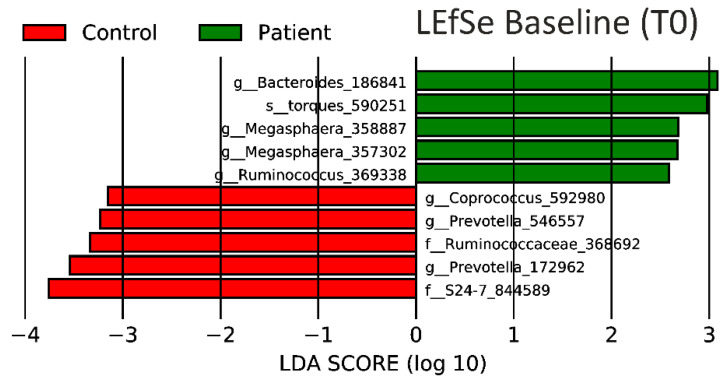
Linear discriminant analysis effect size (LEfSe) results before isoflavone intervention in the control (red) and the patient (green) groups. This analysis indicates the contribution of every strain to the PCOS phenotype. Letters, f (amily), g (enus), or s (train), at the beginning of the name stand for the most specifically known scientific classification of this strain.

**Figure 7 nutrients-12-01622-f007:**
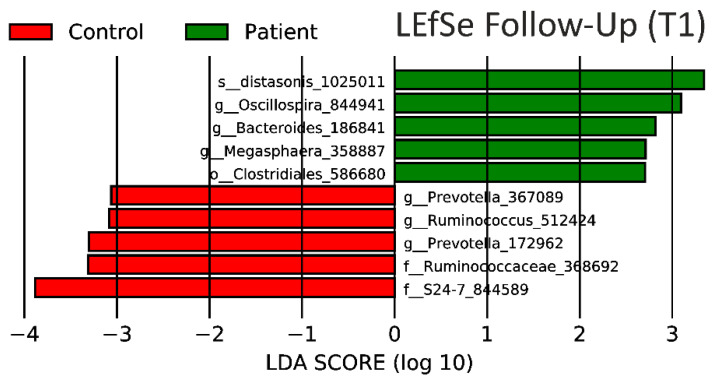
LEfSe results after short-term isoflavone intervention. There is only a marginal change in LDA score of operational taxonomic unit (OTU) in the control group T0 vs. T1 in contrast to PCOS patients, where every OTU has been found to be different except for an *Oscillospira* strain (186841).

**Table 1 nutrients-12-01622-t001:** Anthropometric, metabolic, and steroid hormone parameters in women with polycystic ovary syndrome (PCOS) and control women before isoflavone intervention.

	Reference Range	Controls (*n* = 20)		PCOS (*n* = 24)		
		Median	IQR	Median	IQR	*p*-Value
**Age**		**32**	**12.0**	**27**	**5.9**	**0.003 ****
**Anthropometric parameters**						
Body mass index	18.5–25.0 #	22.3	4.10	24.9	11.75	0.147
Waist to hip ratio	<0.85 #	0.80	0.06	0.82	0.077	0.439
**Metabolic parameters**						
Fasting glucose (mmol/L)	<7.0 †	4.5	0.50	4.7	0.59	0.209
AUC glucose (mmolh/L)	§	10.2	4.52	10.9	3.61	0.273
Fasting insulin (pmol/L)	20.9–173.8	41.4	51.08	84.4	55.25	0.022 *
AUC insulin (mmolh/L)	§	353	427.3	691	562.0	0.009 **
HOMA2-IR	<2.0	0.8	1.05	1.7	1.20	0.027 *
Total cholesterol (mmol/L)	<5.2	4.6	0.64	4.5	1.13	0.699
LDL cholesterol (mmol/L)	<3.4	2.3	0.83	2.3	1.14	0.144
HDL cholesterol (mmol/L)	>1.0	2.0	0.42	1.7	0.49	0.006 **
Triglycerides (mmol/L)	<1.65	0.59	0.25	0.74	0.24	0.010 *
**Serum sex hormones**						
FSH (IU/L)	0.5–61.2 ‡	9.2	8.11	7.5	2.73	0.178
LH (IU/L)	2.0–22.0 ‡	5.8	9.34	9.3	8.60	0.042 *
LH:FSH ratio	§	1.2	1.19	1.5	1.06	0.035 *
SHBG (nmol/L)	18–144	78	29.40	43	41.50	<0.001 ***
AMH (pmol/L)	1.4–65.2	26.8	22.42	61.1	52.59	0.0002 ***
Total testosterone (nmol/L)	0.37–2.1	1.1	0.56	1.3	0.77	0.002 **
DHT (nmol/L)	§	0.34	0.24	0.46	0.528	0.096
Androstenedione (nmol/L)	0.89–7.5	2.6	1.61	4.2	2.69	0.0003 ***
DHEA (nmol/L)	§	13.7	11.37	21.4	12.40	0.015*
DHEAS (µmol/L)	§	3.3	3.74	4.9	2.35	0.073
Free testosterone (pmol/L)	§	10.6	5.86	20.9	13.00	<0.0001 ***
Free DHT (pmol/L)	§	1.3	1.03	3.0	2.19	<0.0001 ***
**PCOS assessment**		**# of cases**	**% of cases**	**# of cases**	**% of cases**	***p*-value**
Oligo-/amenorrhea		1	5	17	71	<0.0001 ***
Hirsutism		1	5	11	46	0.003 **
PCOM		0	0	22	96	<0.0001 ***
**Dietary assessment**		**Median**	**IQR**	**Median**	**IQR**	***p*-value**
Total points		69	27.0	68	31.8	0.318
Grains (% of total points)		19	9.4	17	8.9	0.025 *
Dairy (% of total points)		13	8.6	13	10.2	0.502
Meat/fish (% of total points)		6	6.4	7	6.8	0.649
Fruits/vegetables (% of total points)		17	11.4	19	12.6	0.856
Fats (% of total points)		10	6.0	12	5.2	0.258
Soy (% of total points)		1	1.9	1	2.5	0.658

IQR, interquartile range; AUC, area under the curve; HOMA2-IR, homeostasis model assessment for insulin resistance; LDL, low-density lipoprotein; HDL, high-density lipoprotein; FSH, follicle-stimulating hormone; LH, luteinizing hormone; AMH, anti-Müllerian hormone; DHT, dihydrotestosterone; DHEA, dehydroepiandrosterone; SHBG, sex hormone-binding globulin; DHEAS, DHEA sulfate; PCOM, polycystic ovarian morphology. # as defined by the World Health Organization; † as defined by the American Diabetes Association; ‡ depending on menstrual cycle phase; § reference range not defined. Normally distributed data were compared using unpaired Student’s *t*-tests. Non-normally distributed data were either log-transformed, followed by parametric testing, or compared using Mann–Whitney *U* tests. Categorical data were compared using Fisher’s Exact tests. * *p* < 0.05; ** *p* < 0.01; *** *p* < 0.001. Adapted from [74].

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
