# Peer review of "Impact of Short-Term Isoflavone Intervention in Polycystic Ovary Syndrome (PCOS) Patients on Microbiota Composition and Metagenomics"

_nutrients, 2020, doi:10.3390/nu12061622_

Round 1

Reviewer 1 Report

Haudum et al aimed to investigate isoflavone intervention via soy milk as a potential therapeutic approach in PCOS. To reach this aim they enrolled 24 PCOS and 20 controls that were supplemented for a short time with  isoflavone. They concluded that the administration of isoflavone couls represent a potential therapeutical tool for the management of PCOS. I've raised several issues that need to be solved:

1) The main issue of this manuscript was the sample size that was very small. I suggest the authors to report the calculations of sample size;

2) The PCOS women were insulin sensitive since the median of HOMA-IR was 1.7 ( normal values of HOMA index < 2.5). So, how the authors can state that the administration of isoflavone improve insulin resistance in PCOS women if they had already normal insulin sensititivy and normal fasting glucose? I suggest to provide also informations on the effect of isoflavone in the PCOS women that were insulin resistant (HOMA-IR>2.5). Furthermore, Romualdi et al [ref 14] carried out performed a pilot study supplementing PCOS women with isoflavone for 6 months without finding any improvement of glycoinsulinemic parameters. How the authors discuss this discrepancy with their findings?

3) As well known insulin resistance takes months to change. How the authors could explain a rapid change after isoflavone?

4) Lifestyle is very important in the management of PCOS. Physical activity is also one of the main determinant of insulin sensitivity and it has been reported that could potentiate the effect of isoflavone on insulin resistance (PMID:20539245 ). I suggest the authors to report the data on that. If they do not have data on physical activity, they should report this as limit of the study and also report in the discussion the potential role of physical activity in the improvement of metabolic parameters in PCOS and the potential additive effect on isoflavone treatment. For staters, see PMID 20216276, PMID: 24126551 and PMID: 32229703 

5) The authors should report the effect of isoflavone on lipid profile and provide also comments on discussion on the importance of dyslipidemia in PCOS setting. For starters, see PMID: 29142444, PMID: 24945113, PMID: 27886515 

6) Please report a table reporting the same parameters of Table 1 after isoflavone supplementation in both groups (PCOS and controls) reporting also the statistical differences compared to baseline and between the groups

Author Response

Dear Reviewer!
Thank you very much for your comments!
Please see the attachment.

Best wishes

Christoph Haudum

Reviewer 2 Report

Reviewer comments on Nutrients manuscript number- 797377

Article: Impact of short-term isoflavone intervention in polycystic ovary syndrome (PCOS) patients on microbiota composition and metagenomics

The presence of equol, an isoflavone metabolite produced by intestinal bacteria that acts as a selective estrogen receptor modulator was measured in the urine of women with and without PCOS as phytoestrogens may influence hormone and metabolic signaling. After isoflavone was administered via soy milk, microbiome, metabolic, and predicted metagenome analyses were conducted and baseline stool metagenomic pathways, microbial alpha diversity, and glucose homeostasis were studied. Larger equol production was associated with lower androgen and fertility markers and the importance of phytohormones on PCOS characteristics were determined and are to be studied for a potential therapeutic approach.

General Comments:

Interesting and potentially of great clinical importance, but difficulties in understanding hypothesis, reason for exclusion/inclusion criteria for the specific study are issues that remain. I would be happy to re-review after major reformatting of this paper.

Line 21: State what type of study this is. Interventional, observational, or randomized controlled trial?

My specific comments are provided below.

Please find below an enumerated list of comments and questions based on my review of the manuscript:

Introduction:

Line 43: Add “increased risk of cardiovascular disease” instead of increased cardiovascular disease. There is a need to be explicit.

Line 46-47: Rephrase and fix grammar/punctuation errors. The intention of this sentence is unclear.

Line 51: Briefly define isoflavones to introduce their role to the readers.

Line 54: Please add a reference.

Line 58: Be specific about what exact effects or state that may be uncharacterized.

Line 62: What is the significance of increased calcium levels particularly in relation to PCOS or outcomes of the study? This seems like an estrogen agonist, if it is a SERM. Comment on whether effects are different in different tissues or whether this generally only has agonist activity.

Line 63: What is the significance of decreased incretin levels particularly in relation to PCOS or outcomes of the study?

Line 67-68: Please add a reference.

Line 74: Shifting between equol and isoflavones—would the authors apply any insights from equol to all isoflavones?

Line 76: Note the difference between pre and probiotics for the readers. Add a brief example.

Methods:

Line 84-88: Please add a reference for PCOS criteria. Refer to most recent Rotterdam criteria: https://www.racgp.org.au/afp/2012/october/polycystic-ovary-syndrome/.

Line 89: Clarify inclusion criteria and briefly include criteria even if it was referenced earlier to help with readability.

Line 91: This should be addressed in this discussion section. Please comment on differences between PCOS phenotypes that do not have androgen excess. How would any hormone related interventions help in those without clinical or biochemical androgen excess?

Line 96: Can authors describe the purpose of these inclusion and exclusion criteria on their hypothesis and objective? 

Line 98: Consider moving table 1 to results section.

Table 1: p-value – what is the purpose of sharing p-values? Please refer to: https://www.ncbi.nlm.nih.gov/pmc/articles/PMC4877414/.

Figure 1: move to results section.

Line 213-214: Briefly describe FFQ and add reference. What is the importance of the FFQ in the context of this study?

Line 228: use p-values in table 1 through STROBE guidelines. https://www.equator-network.org/reporting-guidelines/strobe/.

Line 238: Move description of methods to methods sections i.e. timing of isoflavone measurements.

Please add the following in the study design of methods or wherever you see fit.

Add: How was PCOS determined/diagnosed? Self-assessment or by a physician?

How was recruitment conducted?

Results:

Line 225-226: What is the significance of this difference between women with and without PCOS in terms of high TG and low HDL? Please mention in discussion section.

Line 263: Regarding inclusion criteria with only pre-menopausal women, is the age similar among those women? Please clarify age inclusion criteria and provide age ranges.

Line 266: Is this significance of LH:FSH ratio mentioned in discussion? Please elaborate on the significance in the discussion section.

Line 308: Please mention some of these strains and significance in introduction and integrate it into discussion.

Discussion:

Line 338: Refer to figure or table numbers in the discussion.

Line 353: Move reference to the end of the sentence.

Line 354-355:  Rephrase lifetime phases. What do you mean by lifetime phase, please elaborate.

Line 376: Move reference to the end of the sentence.

Line 378-379: Elaborate on the difference between prebiotic and probiotic and why either are significant here.

Line 387: Move reference to the end of the sentence.

Line 412: Write out the acronym KEGG when you mention it for the first time.

Line 426: Move reference to the end of the sentence.

Author Response

(The authors gave the same response as above.)

Round 2

Reviewer 1 Report

The reviewers satisfactorily solved the raised issues.

Author Response

Thanks